# A New Disease Concept in the Age of Processed Foods—Phosphorus-Burden Disease; including CKD–MBD Concrete Analysis and the Way to Solution

**DOI:** 10.3390/nu13082874

**Published:** 2021-08-21

**Authors:** Keizo Nishime, Noriko Sugiyama, Koichi Okada

**Affiliations:** 1Department of Nephrology, Rakuwakai Nijo Ekimae Clinic, 3 Higashitoganoo-cho, Nishinokyo, Nakagyo Ward, Kyoto City 604-8414, Japan; 2Department of Medical Information, Rakuwakai Headquarters, 51-7 Otowa Hachinotubo, Yamashina Ward, Kyoto City 607-8064, Japan; sugiyama-noriko@rakuwa.or.jp; 3Department of Nephrology and Dialysis Surgery, Rakuwakai Otowa Memorial Hospital, 29-1 Otowachinjyucho, Yamashina Ward, Kyoto City 607-8116, Japan; okada-koichi@rakuwa.or.jp

**Keywords:** CKD–MBD, phosphorus burden, processed foods, phosphorus additives, phosphorus binders, fibroblast growth factor 23(FGF23), phosphorus absorption inhibitor, sodium-hydrogen exchanger 3 inhibitor (NaHE3 Inhibitor)

## Abstract

In 2012, the Japanese Society for Dialysis Therapy (JSDT) established the order of correction of P, corrected Ca (cCa), and whole PTH (w-PTH) in the treatment of Chronic Kidney Disease-Metabolic Bone Disorder (CKD-MBD) as P-first. However, there is no report that analyzes whether this rule is in line with reality and what the adequate rate of P is. Therefore, we analyzed the test values of our 48 patients during the year of 2019 and examined the validity of the results. The results showed that the adequate range rates were 70.8% for P, 100% for cCa, and 89.6% for w-PTH. This result is better than the JSDT Web-based Analysis of Dialysis Data Archives (WADDA) P adequacy rate of 66.2%. Although the guideline is P-first, it is often the case that we cannot reach the adequate level; therefore, healthcare professionals and patients often blame each other. We believe that this is due to the mismatch between the modern era of processed foods covered with P additives and treatment methods (P intake restriction and P-binders). The development of processed foods with P additives has brought light and darkness to mankind. The light side is freedom from starvation, and the dark side is a new condition caused by P burden: P burden disease including CKD-MBD.

## 1. Introduction

Cancer and vascular diseases are the two leading causes of death in Japan [1]. Vascular disease has been found to be caused by dyslipidemia and hyperphosphatemia. The former can be controlled by statins [2], while there are still no appropriate means for the latter. Drug therapy for hyperphosphatemia is also unsatisfactory [3]. On the other hand, secondary calciprotein particles (CPPs), which are crystals of Fetuin-A bound to calcium phosphate in blood, have been found to cause vascular damage and calcific masses in soft tissues. The control of phosphorus is becoming increasingly important [4]. The current treatment for hyperphosphatemia consists of limiting P intake from foods and prescribing P-binders. In 2012, JSDT issued guidelines for the treatment of CKD-MBD. In the guideline, P-first was proposed as the order of correction for mineral and bone disorder. However, today, it has not been examined whether this is reasonable or whether P control can be achieved. We examined the concrete analysis with materials from our own clinic.

## 2. Materials and Methods

We elucidate 48 cases of maintenance dialysis at Nijo Ekimae Clinic, Kyoto City, Japan. The data period covered is January to December 2019, with 12 monthly measurements. The analyzed items are P, cCa, and w-PTH. The adequate ranges by JSDT guidelines are as follows. P (3.5–6.0 mg/dL), cCa (8.5–10.0 mg/dL), and w-PTH (35–150 pg/mL). We used statistical software BellCurve for Excel (Social Survey Research Information Co., Ltd., Tokyo, Japan).

## 3. Results

It is very characteristic for the P value to fluctuate, regardless of the treatment of phosphorus binders. The cause of them may be dependent on P uptake from processed foods. More than 90% of the phosphorus added to processed foods is absorbed, indicating that serum phosphorus is greatly affected by the occasional consumption of P additives. (Figure 1)

Even with considerable effort, only 70.8% were able to reach the proper value for P. However, this value is better than 66.2% of JSDT WADDA (Figure 2).

There is very little variation in cCa levels through the year, because we used the Ca sensing receptor enhancers (CaSREs) (Evocalcet or Etelcalcetide) to keep w-PTH at a reasonable level. It is essentially a combination of active vitamin D_3_ (Figure 3).

The attainment of the adequate value of cCa is almost 100%, which is a result of making full use of the CaSREs (Figure 4).

w-PTH is relatively easy to control by CaSREs (Evocalcet or Etelcalcetide) and active VitD_3_ (Figure 5).

Approximately 90% of parathyroid hormones are regulated in the appropriate range. The patient is well controlled with CaSREs (Evocalcet or Etelcalcetide) and active vitamin D_3_ (Figure 6).

The tissue deposition of calcium phosphate is more likely to occur when P x cCa reaches 60 (mg/dL)_2_. We are managing them 16.7% better than JSDT (Figure 7).

P-binders are administered in 52% of patients with sucroferric oxyhydroxide and 40% with lanthanum carbonate hydrate. Precipitated calcium carbonate is prescribed for P-binding and against cCa lowering action by CaSREs. Maintaining normal turnover bone by CaSREs, P and Ca mobilize to the bone matrix, which reduces serum P. Currently, 60% of patients are prescribed more than two binders and CaSREs (Figure 8).

Another key to managing serum phosphorus is to properly manage bone remodeling and mobilize calcium phosphate to the bone. Phosphorus and w-PTH are correlated, so managing at the lower end of the appropriate range for w-PTH can lower phosphorus (Figure 9).

## 4. Discussion

It was indicated 30 years ago that a high P diet causes hyperparathyroidism [5]. The mechanism of this clinico-epidemiological fact has been elucidated over time. FGF23 (Phosphatonin) inhibits 1α-Hydroxylase in the proximal tubule, which prevents the formation of active VitD3 (1α,25(OH)_2_VitD3) and absorption of calcium from the small intestine. As a result, hypocalcemia occurs and parathyroid hormone (PTH) is stimulated, causing hyperparathyroidism. Finally, it was found that PTH causes excessive mobilization of P and calcium from bone, resulting in vascular calcification and ectopic mass calcification [6]. On the other hand, in patients with renal dysfunction, the dysregulation of the αKlotho-FGF-receptor prevents the signaling of the P diuretic factor FGF23, resulting in hyperphosphatemia and vascular calcification.

In 1998, Foley, R.N. et al. reported that mortality from cardiovascular complications in dialysis patients was 10–20 times higher than in the general population [7].

Two years later, Jono, S. et al. tested in vitro whether extracellular high inorganic P causes vascular calcification in human aortic smooth muscle cells (HSMCs). HSMCs were found to be transformed into osteoblasts upon exposure to high inorganic P, producing osteocalcin and Cbfa-1. This effect of high inorganic P was inhibited by phosphonoformic acid. The transmitter of high inorganic P was identified by PCR as the inorganic P transporter-1 (PiT-1). In other words, hyperphosphatemia was thought to cause the conversion of HSMCs to osteoblasts by PiT-1 and active de novo calcification [8]. The results of this experiment correlate with subsequent clinical reports of hyperphosphatemia. Inorganic P loading causes increased cardiovascular mortality in the general population [9], then increased myocardial infarction and subsequent mortality in patients with chronic kidney disease [10], and finally, in chronic hemodialysis patients, which may well explain the increased mortality from cardiovascular complications [11,12].

Hyperphosphatemia causes PiT-1-induced transformation of vascular smooth muscle cells (VSMCs) in renal small arteries into osteoblasts. The resulting vascular calcification of the tunica media of small arteries leads to reduced blood flow and impaired renal function. On the other hand, FGF23 was isolated from neoplastic osteomalacia and was found to cause a decrease in serum P and reduce active VitD_3_ (1α, 25(OH)2D_3_) [13]. Furthermore, αKlotho protein was identified, and FGF23 was found to bind to an αKlotho-FGF coreceptor in the proximal tubule, causing phosphorus excretion and inactivation of 1α-hydroxylase [14]. These findings indicate that, in renal failure, αKlotho dysregulation leads to the inability of FGF23 to function as a P diuretic factor (Phosphatonin), resulting in hyperphosphatemia and vascular calcification.

On the other hand, reports of improvement of vascular calcification by spironolactone continued. Spironolactone was found to inhibit PiT-1-dependent vascular calcification in low Klotho rodents. The mechanism is that spironolactone causes downregulation of PiT-1 expression by aldosterone, which inhibits P absorption and suppresses vascular calcification [15,16].

Furthermore, spironolactone was found to promote the expression of proximal tubular αKlotho by upregulating 1α-hydroxylase [17]. Unexpectedly, spironolactone was found to dramatically reduce cardiovascular and cerebrovascular morbidity and mortality in hemodialysis patients: in a three year follow-up study of 157 patients, hospitalization for cerebral cardiac events was 5.7% in the spironolactone-treated group versus 12.5% in the control group, and all-cause mortality was reported to be drastically reduced to 6.4% in the treatment group versus 19.7% in the control group [18].

Treatments for hyperphosphatemia are listed in Figure 10.

In the age of processed foods that contain large amounts of P, it is impossible to eliminate P by using P-binders. P absorption inhibitors are long awaited. In America, a P absorption inhibitor will soon be in use, and its effectiveness is valid in clinical trials.

A great deal of effort is being made to reduce phosphorus. First, it is recommended to reduce P intake (a. consuming plant proteins with low P availability (40%); b. selecting proteins with low P content), reducing additives in foods and drugs containing inorganic P, 90% of which is absorbed [19,20,21,22,23]. Second, it is recommended to prescribe P-binders. Unfortunately, we can only eliminate a portion of the large amount of P additives that come in during the age of the processed foods.

Third, in near future, there is the option of P absorption inhibitors at the intestinal level (sodium-hydrogen exchanger 3 inhibitor) [24]. Incidentally, Tenapanor hydrochloride, which inhibits P at the intestinal level even when large amounts of P additives are introduced, has been reported to lower serum P by 2.5 mg/dL (Figure 11) [25].

The ultimate means of preventing P in the age of processed foods involves P absorption inhibitors rather than binders. Large amounts of P additives as preservatives, seasoning agents, and colorants have been introduced into the body. There will be a need for drugs that inhibit the absorption of P at the level of the small intestine. During low P uptake, transcellular active absorption works using a NaP cotransporter. When During massive P uptake, paracellular passive permeability works. NaHE 3 Inhibitor (Tenapanor) suppresses P absorption by tightening the tight junction [26].

Dietary guidance to reduce P intake is commonly practiced but is the least effective. Neither patients nor medical staff expect significant results and are often at odds with each other [27]. Unfortunately, P-binders have not been effective enough to be evaluated from a cost–benefit perspective. In our analysis, despite the use of adequate P-binders, P within the adequate range was only 70.8%. Moreover, the national data (66.2%) from JSDT WADDA (2019) shows even worse results than ours. In fact, as we show in our graph, the P level fluctuates greatly from month to month, even though the amount of P-binders medication remains constant; we believe that the intake of processed P-rich foods is more than the amount of P binded. In America, P intake was 470 mg/day in the 1990s, but in recent years, it has exceeded 1000 mg/day due to the widespread use of processed and fast foods [20]. Phosphorus levels (>5.5 mg/dL) in the U.S. are worsening every year, reaching 42% at the end of 2020 [28]. However, it has been indicated that a more important matter is the lack of literacy about processed food centered diets among health care providers and patients [29,30]. A survey of medical and nursing students in Japan reported a lack of knowledge about P additives in processed foods and soft drinks [31], while more than 50% of patients at a dialysis clinic in the UK believed that dietary restrictions were not necessary if they were taking P-binders [20]. Protein and P content are twice as high in processed meats as in raw meats due to food additives; the P to protein ratio is similarly high, and the dangers of inorganic P additives are not well recognized [19]. Inorganic P additives can be reduced to 51% for vegetable-derived P, 40% for bean-derived P, 38% for meat-derived P, 70% for flour-derived P, and 19% for cheese by cooking such as boiling [20]. However, what is fundamental is the regulation of P additives in processed foods by national and international organizations and the mandatory labeling of P content in products [19,32,33].

One typical case always showed P levels above around 12 mg/dL in the outpatient setting, but when admitted to the hospital, the P levels were almost normal (Figure 12).

This case shows approximately 12 mg/dL of P throughout the year. However, after being hospitalized and placed on a P-restricted diet, the P dropped to almost the adequate range. Her hyperphosphatemia was originated in the patient’s diet (Figure 13).

Medical staff including physicians have criticized patients for not following P-restricted diets and not taking P-binders. Recently, Block, G.A. has been raising concerns about” the erroneous pejorative labeling of patients as non-compliant when their serum P fails to conform to clinical expectations despite prescribed P-restricted dietary limits and P binders” [25].

He notes that current P-binding agents have limitations in removing P and that there is a need for new drugs, or absorption inhibitors.

In other words, the P-restricted diet in the hospital brings the level to normal. This shows how today’s inexpensive processed foods, especially fast foods, are covered with inorganic P additives. In addition, not many socio-economic classes can afford to buy raw materials and eliminate P from their diets due to the economic reason [34].

In the age of processed foods that contain large amounts of P, it is impossible to eliminate P by using P-binders. Therefore, P absorption inhibitors are long awaited. P absorption inhibitors are nearly in use in the U.S., and their effects in clinical trials are as expected. In addition, as an international issue, regulation of P as an additive and mandatory labeling of P content in products are required. While inexpensive processed foods have enabled humanity to escape starvation in times of disaster, large amounts of P added as preservatives, seasoning agents, and colorants have entered the body, causing vascular calcification throughout the body. Currently, there are temporary P-binders, but in the future, there will be a need for drugs that inhibit the absorption of P at the level of the small intestine. There is a need to prevent P from the stage of apparently healthy people.

Kuro-o, M. insists that we should take FGF23 as a marker of P burden. Even when serum P is not increased, P burden is identified by elevated FGF23 [35].

## 5. Conclusions

We suggest that P additives in processed foods are the root of the problem that has led to today’s proliferation of hyperphosphatemia cases, and that there is an urgent need to regulate P additives in processed foods and to label the amount of P added to foods. While mankind is being freed from starvation by the development of inexpensive processed foods, CKD-MBD has been brought as a new disease caused by high P ingredients. In the age of processed foods, large amounts of P flow into the human body. Two major strategies exist: one is limiting P intake from foods, and another is the prescription of P-binders. There are insufficient results resolving hyperphosphatemia. Today, the rational response to hyperphosphatemia is to develop P absorption inhibitors to prevent the large amount of P introduced to the body at the intestinal level. In addition, the use of FGF23 as a marker for early detection of P burden in healthy people is essential, even if their P levels are apparently normal.

## Figures and Tables

**Figure 1 nutrients-13-02874-f001:**
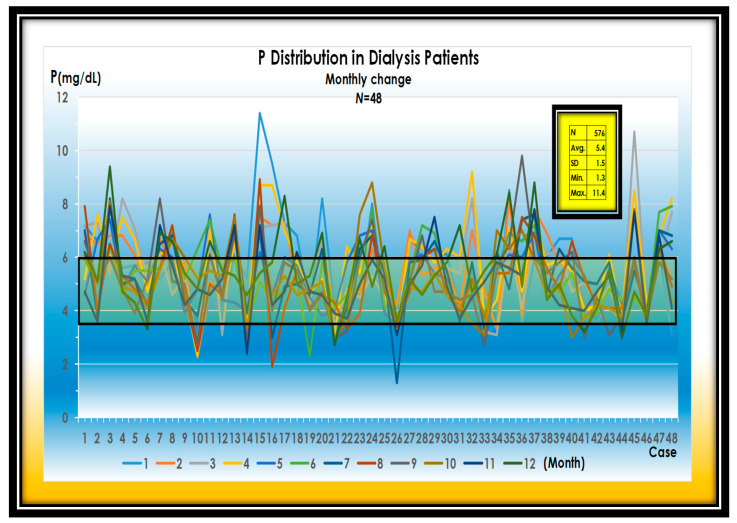
Changes in serum P levels in 48 dialysis patients. Average of P is 5.4 mg. SD is 1.5 mg/dL. Maximum of P is 11.4 mg/dL.

**Figure 2 nutrients-13-02874-f002:**
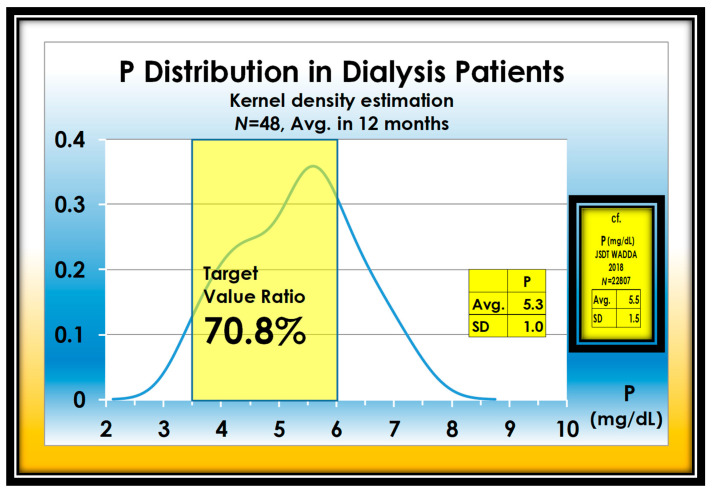
Adequate ratio of serum P in 48 dialysis patients is 70.8%.

**Figure 3 nutrients-13-02874-f003:**
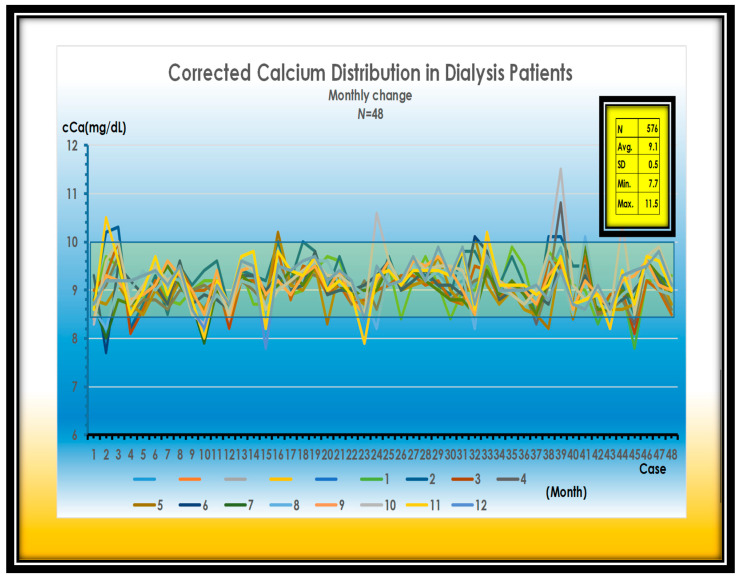
Changes in cCa levels in 48 dialysis patients. Average of cCa is 9.1 mg. SD is 0.5 mg/dL. Maximum of cCa is 11.5 mg/dL.

**Figure 4 nutrients-13-02874-f004:**
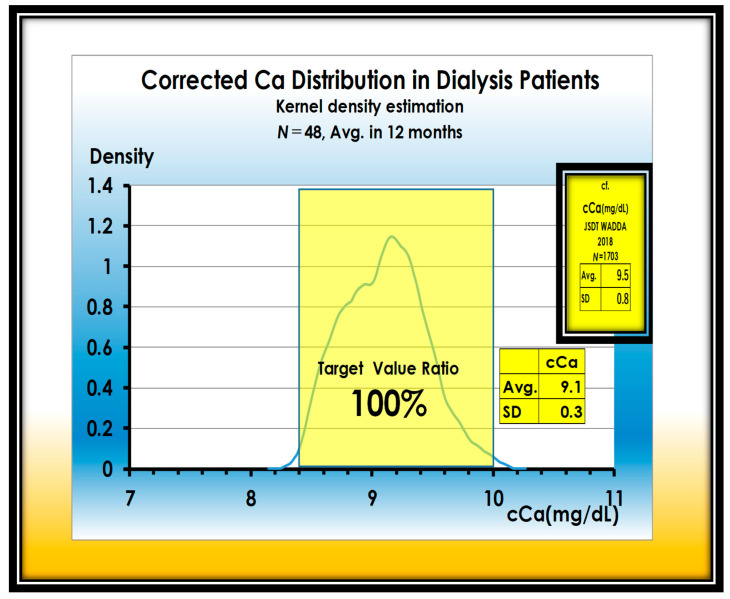
Adequate ratio of cCa in 48 dialysis patients is 100%.

**Figure 5 nutrients-13-02874-f005:**
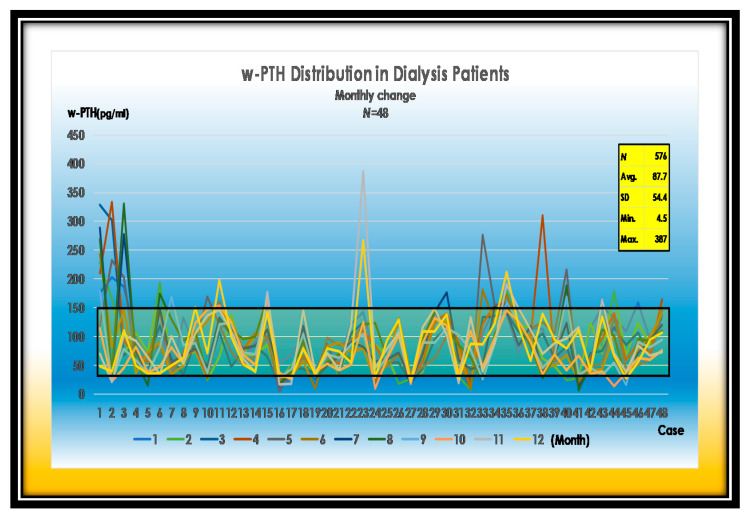
Changes in w-PTH levels in 48 dialysis patients. Average of w-PTH is 87.7 pg/mL. SD is 54.4 pg/mL. Maximum of w-PTH is 387 pg/mL.

**Figure 6 nutrients-13-02874-f006:**
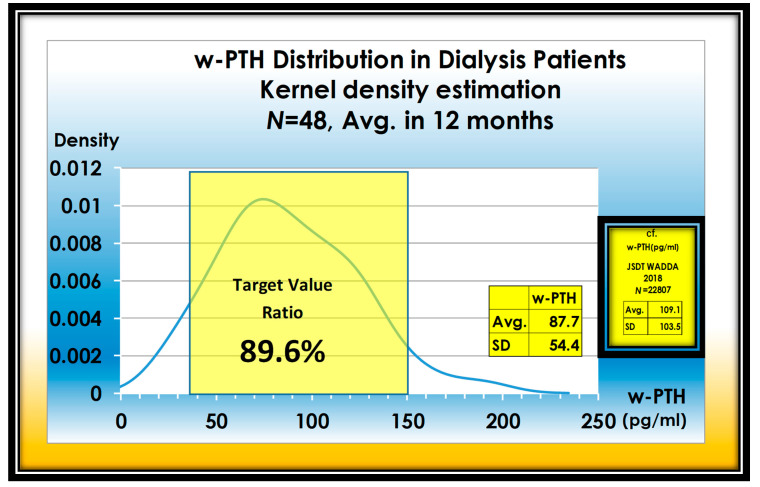
Adequate ratio of w-PTH in 48 dialysis patients. Adequate range ratio of w-PTH reaches 89.6%.

**Figure 7 nutrients-13-02874-f007:**
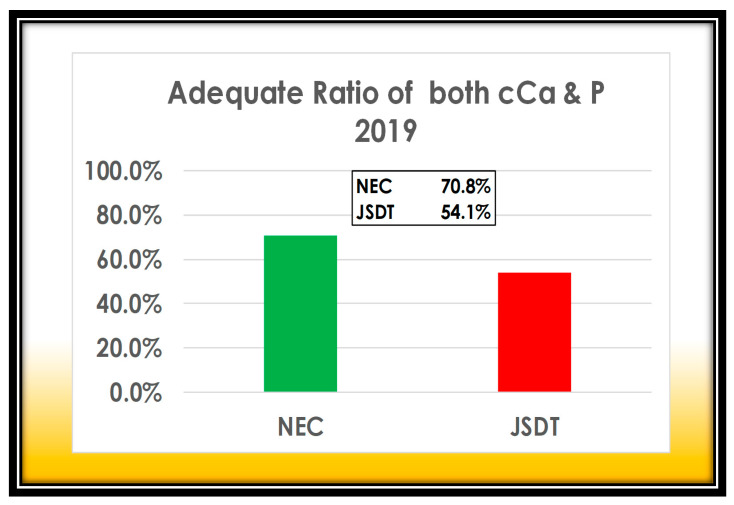
Simultaneous adequate range ratio of P and cCa in our cases comparing to JSDT. Our result is 16.7% superior to 54.1% of JSDT WADDA. NEC; Nijo Ekimae Clinic.

**Figure 8 nutrients-13-02874-f008:**
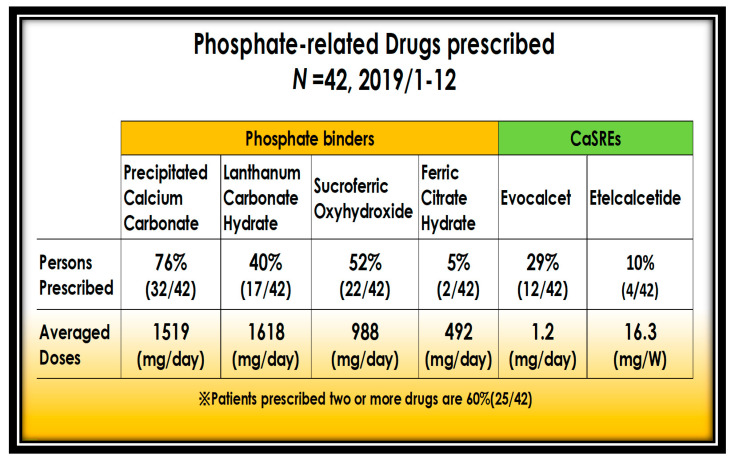
Use of P-binders and CSREs.

**Figure 9 nutrients-13-02874-f009:**
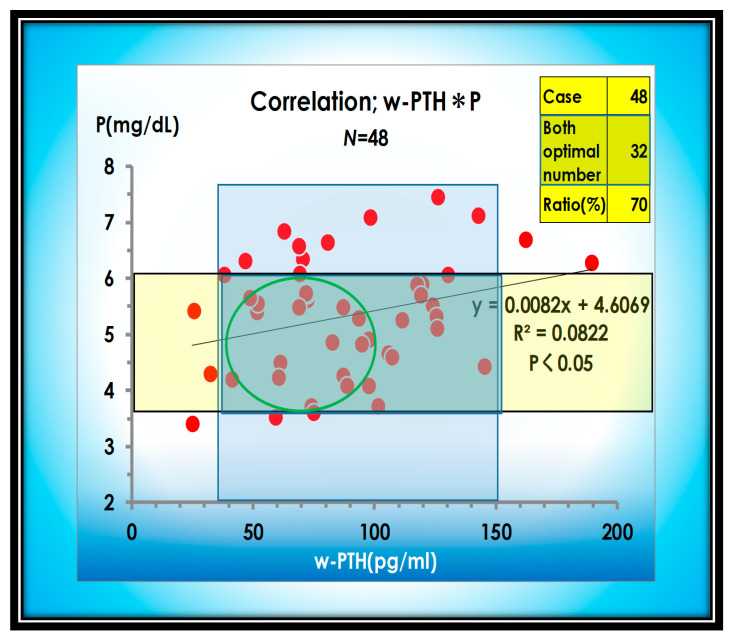
Correlation between P and w-PTH. P and w-PTH are positively correlated.

**Figure 10 nutrients-13-02874-f010:**
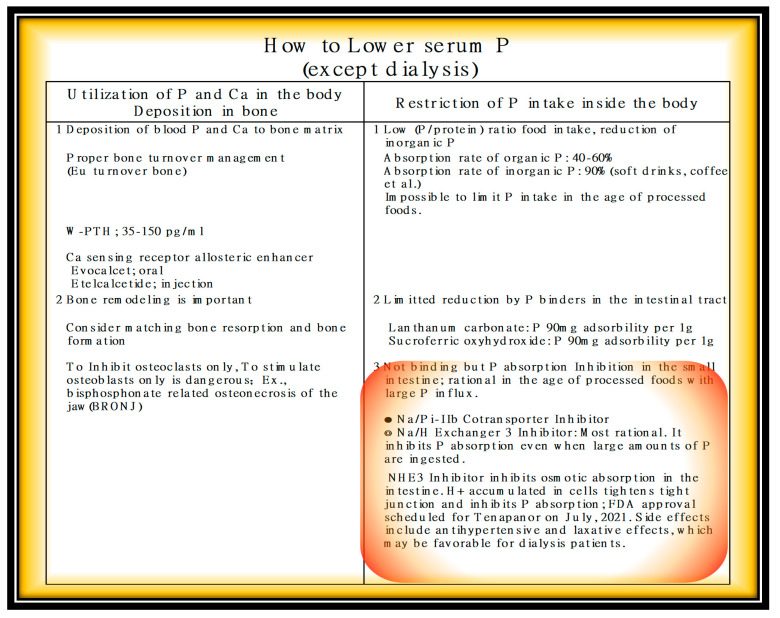
How to lower serum P (except dialysis).

**Figure 11 nutrients-13-02874-f011:**
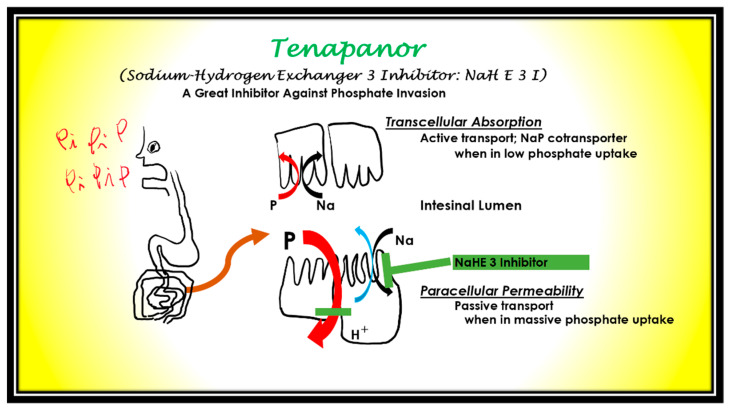
The mechanism of NaHE 3 Inhibitor. NaHE 3 Inhibitor; sodium hydrogen exchanger inhibitor.

**Figure 12 nutrients-13-02874-f012:**
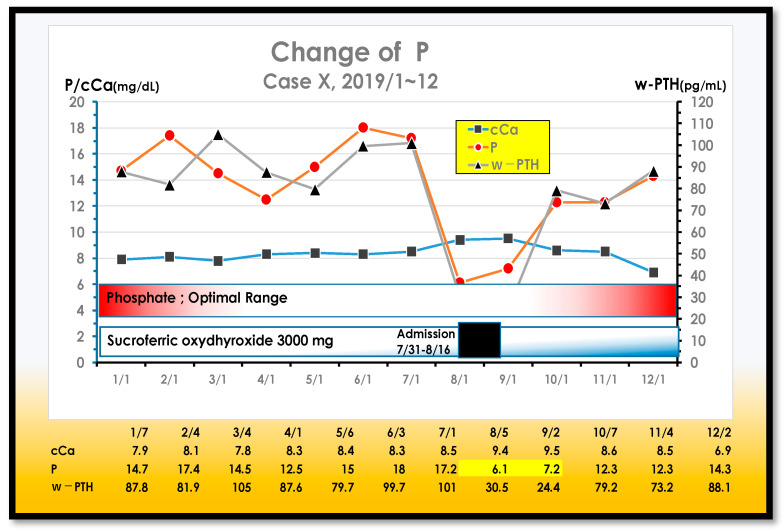
Case X always showed hyperphosphatemia.

**Figure 13 nutrients-13-02874-f013:**
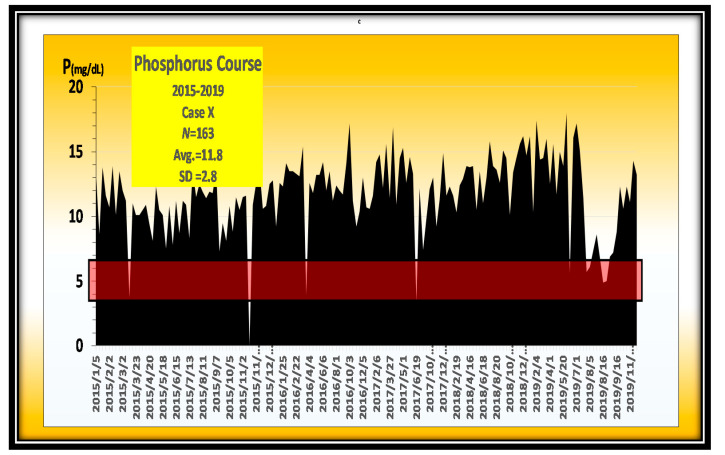
Changes in P over 5 years in a typical hyperphosphatemia case. The P is always around 12 mg/dL during 5 years.

## Data Availability

Provided upon request.

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
