# Peer review of "A New Disease Concept in the Age of Processed Foods—Phosphorus-Burden Disease; including CKD–MBD Concrete Analysis and the Way to Solution"

_nutrients, 2021, doi:10.3390/nu13082874_

Round 1

Reviewer 1 Report

Major point

  1. It is thought that the subject of this study provides a direction on which of the phosphorus-restricted diet or the use of phosphorus-binders is more important in current CKD-MBD. However, it is not representative that this study suggests simply calcium, phosphate, and PTH levels of a small number of hemodialysis patients from a single center, comparing the JSDT guideline.
  2. The content of the introduction is not sufficient.
  3. The main subject in this study is related to the regulation of phosphorus, but the sudden description of spironolactone in the discussion section hinders the unity of the text. Although spironolactone was presented to refer to the treatment of vascular calcification and cardiovascular disease caused by hyperphosphatemia, it is unnecessary.
  4. It is appropriate to report in the form of a case report to refer to the comparison of the p-binder and the p-restricted diet with the case of one patient. The current status of the p-restricted diet and the compliance for p-binder of all patients in this study should be summarized and described in detail.
  5. Please suggest the dose of phosphate-related drug used in the patient groups.

Author Response

1.It is thought that the subject of this study provides a direction on which of the phosphorus-restricted diet or the use of phosphorus-binders is more important in current CKD-MBD. However, it is not representative that this study suggests simply calcium, phosphate, and PTH levels of a small number of hemodialysis patients from a single center, comparing the JSDT guideline.

【Author’s reply】
Thank you for  question 1.

 This paper analyzes the fact that health care professionals and patients are not aware of the enormous amount of phosphorus additives in the age of processed foods and the fact that we are living almost entirely on processed foods. The analysis of 48 cases demonstrates the fallacy that P burden can be solved by simple dietary guidance and the use of P binders that can only partially remove the large amount of inorganic phosphorus entering the human body.   
   Our single center result(optimum P of 70.8 %) is supported by the result of the JSDT (optimum P of 66.2 %) mass data and the US-DOPPS (P 42% over 5.5mg/dL) mass data .The fundamental problem is that the regulations against the influx of large amounts of phosphorus additives into the human body in the era of processed foods and the mandatory labeling of foods have been neglected. 
 On the other hand, the invention of processed foods has the advantage of being inexpensive and can be stored for a long time, therefore there are reports of high phosphorus levels in low-income groups regardless of race. A practical solution is to use phosphorus absorption inhibitors in the small intestine rather than phosphorus binders, which can only remove a portion of the phosphorus. We need to realize the mismatch between the large uncontrollable influx of phosphorus and the current treatment methods.

2.The content of the introduction is not sufficient.

【Author’s reply】
Thank you for  question 2.

I supplemented the literatures and provided additional explanations as shown in red.

 Cancer and vascular diseases are the two leading causes of death in Japan【1】. Vascular disease has been found to be caused by dyslipidemia and hyperphosphatemia. The former can be controlled by statins【2】, while there are still no appropriate means for the latter. Drug therapy for hyperphosphatemia is also unsatisfactory【3】. On the other hand, secondary calciprotein particles (CPPs), which are crystals of Fetuin-A bound to calcium phosphate in blood, have been found to cause vascular damage and calcific masses in soft tissues. The control of phosphorus is becoming increasingly important【4】. The current treatment for hyperphosphatemia consists of limiting P intake from foods and prescribing P-binders. In 2012, JSDT issued guidelines for the treatment of CKD-MBD. In the guideline, P first was proposed as the order of correction for mineral and bone disorders. However, today, it has not been examined whether this is reasonable or whether P control can be achieved. We examined the concrete analysis with materials from our own clinic.

3.The main subject in this study is related to the regulation of phosphorus, but the sudden description of spironolactone in the discussion section hinders the unity of the text. Although spironolactone was presented to refer to the treatment of vascular calcification and cardiovascular disease caused by hyperphosphatemia, it is unnecessary.

【Author’s reply】
Thank you for  question 3.

The description of spironolactone corresponds exactly to the title” Phosphorus-Burden Disease; CKD-MBD-Concrete Analysis and the Way to Solution”. In 2014, Matsumoto's clinical research report showed that the use of spironolactone halved the number of cerebral cardiac events and dramatically reduced the mortality rate in  dialysis patients. The reason for this was found to be the inhibition of PiT1. This is one of the ways to the solution of phosphorus-burden disease, and it is an indispensable description.

4.It is appropriate to report in the form of a case report to refer to the comparison of the p-binder and the p-restricted diet with the case of one patient. The current status of the p-restricted diet and the compliance for p-binder of all patients in this study should be summarized and described in detail.

【Author’s reply】
Thank you for  question 4.

In clinical practice, a typical case clearly shows the nature of the problem. This case illustrates the limitations of the current treatment, which for five years has been ridiculing patients as non-compliant, as pointed out by Block, G.A.
In the end, it is a contradiction between the lack of understanding of the problem in the age of processed foods and the lack of corresponding therapeutic agents. Therefore, this is a necessary presentation.
 At each daily rounds, I explain the phosphorus restriction and confirm that the phosphorus-binders are taken.

5.Please suggest the dose of phosphate-related drug used in the patient groups.

【Author’s reply】
Thank you for  question 5.

Figure 8 provides a summary of the use of phosphorus binders and calcimimetics. Both are formulated as needed. Lanthanum carbonate is 40% (average daily prescription: 1618 mg).Sucroferric oxyhydroxide was prescribed in 52% (average daily prescription 988 mg) and precipitated calcium carbonate in 76% (average daily prescription 1519 mg). Two or more drugs were prescribed to 60% of the patients.

I would like to express my profound appreciation for the reviewers' comments.
                              Keizo Nishime
                                                               Aug.1, 2021

Reviewer 2 Report

It is important to improve the bibliography and make a written review in the introduction.

Author Response

Reviewer 2
It is important to improve the bibliography and make a written review in the introduction.

【Author’s reply】
Thank you for  suggestions.

I followed your instructions to correct the literatures and explain the results.

I would like to express my profound appreciation for the reviewers' comments.
                              Keizo Nishime
                                                               Aug.1, 2021

Round 2

Reviewer 1 Report

It is a well-known fact that vascular calcification is aggravated by the accumulation of P. Even in various guidelines, P-binding agents are used to remove P ingestion because dialysis does not remove all of the ingested P. However, there is a difference in P removal depending on the type and dose of the phosphorus binder, and the recommended dose of the drug is limited. It is recommended to reduce the high P diet in this paper. The authors suggest that the intake of P additives is a way to overcome starvation, and for this reason, it is a logical leap forward to say that vascular calcification is aggravated by P accumulation. Although it may be possible to say that this trend is high, especially in low-income countries, in general, consumption of p additives is not a subsistence intake, and it is not a bigger problem in low-income countries. Junk food consumption is a common phenomenon all over the world, and this article may cause misunderstandings in the socioeconomic field. In addition, it is effective to use drugs that reduce phosphorus absorption, but the author's center does not explain this part enough. Although one case is presented in the discussion section, the situation of each patient will be different, and it is necessary to describe in detail the contents of intake and treatment for P additives. In the discussion section, it is not enough to list the guidelines and various studies already suggested on the regulation of P. For each treatment, how it was applied at the author's center, how vascular calcification was measured, and the actual change when P additives were suppressed should be fully described, and the specificity and limitations of this paper should be presented.

Author Response

Thank you for your second comment.
【Author's reply】
 First of all, what the first and second comments have in common is the concern that the reviewers may not understand the argument of this paper.
This is because there are too many comments that miss the point or demand answers that do not need to be answered.   
 I will state the purpose of this paper for the reviewers. Then, I will give the minimum necessary response.
 In this paper, I argue that the Japanese Society of Dialysis Therapy's P-first guideline for CKD-MBD does not correspond to reality, based on my clinic's test results, the Japanese Society of Dialysis Therapy's national survey, and the US-DOPPS results. Next, I argued that the reason for this is due to our dietary habits centered on processed foods that contain a lot of inorganic phosphorus additives. The invention of inexpensive, long-lasting processed foods that are not affected by climate change has freed humanity from starvation, but the heavy use of inorganic phosphorus additives as preservatives, seasonings, and coloring agents has given rise to a new disease, phosphorus burden disease. 
 The danger of phosphorus overload disease was first shown by Calvo's warning (1990) that a high phosphorus diet causes secondary hyperparathyroidism. Foley, R.N. (1998) reported that chronic kidney disease is 10 to 20 times more likely to cause cardiovascular disease, and Jono, S. (2000) reported that high phosphorus causes the transformation of vascular smooth muscle cells into osteocytes and de novo calcification of small artery tunica media. Kuro-o, M. (1997) discovered the hyperphosphorous premature aging mouse and new hormone Klotho and subsequently reported the elucidation of tissue damage caused by the binding of calcium phosphate to Fetuin-A (Calciprotein particle). Despite the suggestions of many researchers, I think there is a problem with the current situation where the regulation of inorganic phosphorus additives and the mandatory labeling of the amount of inorganic phosphorus in foods are not underwent. And what is even more problematic is that medical professionals and patients are not aware of this point. It is not possible to control phosphorus with a diet that avoids inorganic phosphorus, which is difficult for people of normal economic level, or with partial binders of inorganic phosphorus that flow into the human body in large quantities. As Block, G.A. (2019) has already shown in his paper, the effect of lowering blood phosphorus by 2.5 mg/dL and the laxative effect of residual sodium in the intestines are multiple. effect and antihypertensive effect, among other multiple benefits. Clinical trials are also being conducted in Japan.
 Phosphorus burden disease in the age of processed foods is a new disease concept and a challenge for the entire human race. Even if serum phosphorus is apparently normal, phosphorus burden occurs, and FGF23 should be used as a marker to identify phosphorus burden at an earlier stage, according to Kuro-o, M.
